# High Internal Phase Emulsions Preparation Using Citrus By-Products as Stabilizers

**DOI:** 10.3390/foods11070994

**Published:** 2022-03-29

**Authors:** Joana Martínez-Martí, Amparo Quiles, Gemma Moraga, Empar Llorca, Isabel Hernando

**Affiliations:** Food Microstructure and Chemistry Research Group, Department of Food Technology, Camí de Vera, s/n, Universitat Politècnica de València, 46022 Valencia, Spain; joamarm8@etsiamn.upv.es (J.M.-M.); mquichu@tal.upv.es (A.Q.); gemmoba1@tal.upv.es (G.M.); emllomar@tal.upv.es (E.L.)

**Keywords:** biopolymers, pomace, microstructure, rheology, valorization, emulsifier

## Abstract

The citrus juice industry produces about 50% of by-products. Citrus pomace (CP) contains many polysaccharides (mainly cellulose and pectin), which could act as stabilizers and emulsifiers. The aim of this work was to obtain high internal phase emulsions (HIPEs) using unmodified CP at different concentrations to valorize citrus by-products. The synergic effect of pea protein isolate (PPI) with CP to stabilize the HIPEs was also studied. HIPEs structure was analyzed using rheological and microscopy studies as well as color and physical stability of the emulsions. According to rheological data, all samples exhibited a solid-like behavior, as elastic modulus (G’) was higher than viscous modulus (G’’) within the viscoelastic linear region; as % CP and % PPI increased, greater values of G’ and apparent viscosity (η) were achieved. Microscopic images showed that oil droplets had a polyhedral shape and were enclosed by a thin layer of CP and PPI. Increasing concentrations of CP and PPI enhanced oil droplets packaging. Emulsions’ physical stability was better when adding PPI. The results showed that stable HIPEs with 1.25% of CP and PPI over 0.5% can be obtained. These HIPEs could be used to formulate emulsions for food applications, such as mayonnaises, fillings, or creams.

## 1. Introduction

Citrus is one of the main fruits processed worldwide. Its processing to obtain juice generates large amounts of by-products, about 50% of the fresh fruit weight [1]. This by-product, also known as citrus pomace (CP), comprises skin, pulp, and seed, and among its components, there are natural biomolecules, such as cellulose, hemicellulose, pectin, protein, lignin, soluble sugars, and essential oils, that could be used in the food industry [2]. Cellulose and pectin are the most abundant components, representing 45% and 35% of the total weight basis, respectively. Cellulose is an amphiphilic polysaccharide, which can be adsorbed at the interface of oil droplets, avoiding coalescence because of the coating, thus acting as an emulsifier [3]. Cellulose not adsorbed at the interface is placed in the continuous phase, raising the viscosity of emulsions and forming a network that traps the oil droplets, improving the stability of the emulsion, and therefore acting as a stabilizer [4]. Pectin is used primarily for its gelling and stabilizing abilities. In addition, it has emulsifying properties, which are mainly due to the protein residues present in the pectin [5,6]. Regarding its stabilizing capacity, pectin also raises the viscosity of the emulsion by increasing the interaction between particles and reducing the separation between phases [7]. Other authors [8,9] have used apple pomace as an emulsifier and stabilizer because it has a relevant percentage of pectin (9–15% wt./wt. on dry basis). Using pectin and cellulose as stabilizers is common, but there is no literature about directly using CP with technological purposes. Therefore, one option to valorize citrus pomace could be its use as an emulsifier and stabilizer to obtain emulsions.

Saturated fat provides desirable textural properties to foods, a pleasant mouthfeel when ingested, and extends foods’ shelf life, as it is less susceptible to oxidation than monounsaturated and polyunsaturated fatty acids. They are widely used in the frying, confectionery, bakery, and pastry industries. Consuming saturated fat instead of polyunsaturated fat raises LDL cholesterol levels, and this excess is considered a risk factor for cardiovascular diseases [10]. Institutions and consumers increasingly reject saturated fats, so it is necessary to find healthy alternatives that allow the preservation of the sensory and quality properties of food. The structuring of oils is a promising alternative for developing new products with an improved nutritional profile, with a high content of mono- and polyunsaturated fatty acids, low in saturated fats, and without trans fats [11].

High internal phase emulsions (HIPEs) are structured systems defined by an internal phase volume fraction above 0.74. When the internal phase volume fraction exceeds this value, the dispersed phase droplets are deformed into polyhedral geometries, which are separated by thin films of continuous phase [12]. HIPEs have semi-solid textures; they often behave as elastic solids below a critical stress and as viscous liquids above this stress, and their viscosity decreases as the shear stress increases [13]. The use of HIPEs is being studied for various food applications, such as mayonnaise substitutes or fat replacers in chocolate [14], which opens up the possibility of using them as substitutes for saturated fats. To obtain stable HIPEs, polysaccharides and proteins are used as structuring agents because of their stabilizing and emulsifying properties, respectively [15]. The CP contributes many polysaccharides, such as cellulose and pectin, which in the absence of additional protein could act as emulsifiers and stabilizers and in the presence of added protein could mainly act as stabilizers. Previous studies [16,17,18] using citrus by-products have extracted the pectin in those by-products to obtain emulsions with less than 50% of oil, but to the best of our knowledge, no studies using CP without modifications to obtain HIPEs have been developed. Pea protein is suitable as an emulsifier because of its amphiphilic nature, illustrated by a good balance between its hydrophilicity and hydrophobicity [19]. Besides, pea protein is not a cause of food allergy problems, which makes it appropriate to be incorporated in new food formulations. Therefore, pea protein would be a good option for obtaining more stable HIPEs.

The aim of this work was to obtain HIPEs as potential replacers of saturated fats, using unmodified CP as an emulsifier and stabilizer to valorize citrus by-products. Adding pea protein isolate (PPI) to confer a higher stability to the emulsions was also evaluated. Color, oil loss, rheological behavior, and microstructure characteristics of the HIPEs were evaluated to understand their stability.

## 2. Materials and Methods

### 2.1. Materials

Citrus pomace (moisture 8.3%; carbohydrates 80.5%, of which soluble dietary fiber meant 44.8% and insoluble dietary fiber meant 26.4%; protein 6.68%; fat 1.67%; ash 2.85%) was donated by the Zumos Valencianos del Mediterráneo S.L. company. Pea protein isolate (PPI) (protein 84% db) was supplied by the HSN Network Ltd. Eden Point (Granada, Spain). Sunflower oil was purchased from a local supermarket (Valencia, Spain). Fluorescent dyes, Nile Red, and FITC were supplied by Sigma-Aldrich, Inc. (Saint Louis, MO, USA) and Electron Microscopy Science, Inc. (Hatfield, PA, USA), respectively.

### 2.2. Citrus Pomace Powder Preparation

Citrus pomace was homogenized at 1400 rpm in an emulsifying mincer for 10 min (CUP 8Tr, Eurofred, Barcelona, Spain). The ground pomace was centrifuged (SORVALL SUPER T21, Ramsey, MN, USA) at 15,951× *g*, for 10 min at 15 °C, and then, the supernatant was removed. The solid residue was dried at 40 °C and −0.7 bar in a vacuum oven (Vaciotem-T, J.P. Selecta, Barcelona, Spain) until constant weight. The dried solid residue was ground in a mincer (Moulinette 1,2,3, Moulinex, Alençon, France) and then milled in a cooling water jacketed mill (M20, IKA, Staufen, Germany) at 2 min intervals for 10 min to obtain powder. Citrus pomace powder (CP) was packed under vacuum (EDESA, Granollers, Spain) and stored in a desiccator until use.

### 2.3. Emulsion’s Preparation

Emulsions were prepared with an oil/water ratio of 80:20. Citrus pomace powder (CP) and PPI (if required) were dispersed in distilled water and stirred for 2 h at room temperature. The preparation of emulsions was conducted according to Ruan et al. [20], with slight modifications. The aqueous phase was sheared at 8000 rpm for 30 s using a homogenizer (Ultraturrax T-18, IKA-Werke GmbH & Co. KG, Staufen, Germany). Next, oil was gently added while homogenizing at 14,000 rpm. Finally, emulsion was homogenized at 16,000 rpm for 30 s. Resulting emulsions were stored at 4 °C for analysis after 24 h.

According to the different concentrations of CP and PPI used in the emulsions and determined after preliminary studies, a letter was assigned to each emulsion, as follows in Table 1.

### 2.4. Characterization of Emulsions

#### 2.4.1. Color

The CIE L*a*b* color coordinates, chroma (C*) and hue angle (h*) were measured with a Colorimeter Minolta (Chroma Meter CR-400/410, Konica Minolta, Chiyoda, Japan) equipped with standard light source C and standard observer 2°. For this, an aliquot of emulsions was introduced into a 40 mm (diameter) × 12 mm (height) Petri dish placed on a white background. The total color difference (∆E*) between samples was calculated according to Equation (1), considering sample A as the reference:(1)ΔE*=((ΔL2)+(Δa2)+(Δb2))12

#### 2.4.2. Oil Loss

To determine the oil loss of emulsions, an aliquot of 1 g was placed in an Eppendorf of 1.5 mL and centrifuged (Centrifuge 5415R, Eppendorf, Hamburg, Germany) at 540× *g* for 60 min at 25 °C. After centrifugation, the free oil was removed, and the Eppendorf with the sample was weighed. Oil loss was calculated using Equation (2):(2)$$Oil\;loss\;\left(\%\right)=\left(\left(m_{2}-m_{3}\;\right)\right)/\left(\left(m_{2}-m_{1}\;\right)\right)\times 100$$
where *m*_1_ is the mass of empty Eppendorf, *m*_2_ is the mass of Eppendorf and sample before centrifugation, and *m*_3_ is the mass of Eppendorf and sample after centrifugation and removal of free oil.

#### 2.4.3. Rheology

For the study of the rheological behavior of emulsions, a rotational rheometer (Kinexus Pro+, Malvern Panalytical, Malvern, UK) equipped with a Peltier system for temperature control and a parallel plate geometry of 40 mm diameter was used. The geometry gap was set at 1000 µm, and all measures were performed at 20 °C. Amplitude sweeps (stress = 0.01–100 Pa, frequency = 1 Hz) and frequency sweeps (frequency = 0.1–10 Hz, stress = 0.3 Pa) were conducted, recording elastic (G’) and viscous (G’’) moduli. Flow curves (rate ramp from 0.1 to 20 s^−1^ during 3 min) were conducted to study apparent viscosity (η). All measurements were performed in triplicate.

#### 2.4.4. Microstructure

The microstructure of the emulsions was studied using light field microscopy (LM) and fluorescence microscopy. For this, a Leica microscope (Leica DM5000, Leica, Wetzlar, Germany) was used.

To study the samples by LM, a portion of the sample was deposited on a glass slide and covered with a coverslip. Then, it was observed under the microscope by bright field. To study samples by fluorescence, they were stained using Nile Red (fat-specific staining agent) and FITC (biopolymers-polysaccharides and proteins-staining agent), after depositing the sample on the glass slide. Samples were covered with a coverslip and observed under the microscope using an N21 filter (λex = 515–560 nm, λem = 590 nm) for the Nile Red and a GFP filter (λex = 470/40 nm, λem = 525/50 nm) for the FITC.

All samples were visualized using 10× objective lenses. Images were captured and stored at 1024 × 1024 pixels using the image capture software. The area of the fat globules was measured using ImageJ software (U.S. National Institutes of Health, Bethesda, MD, USA).

### 2.5. Statistical Analysis

The statistical analysis of the results was conducted using the Statgraphics Centurion XVII software (Statgraphics Technologies, Inc., The Plains, VA, USA). A multifactorial analysis of variance (ANOVA) was performed, where the factors analyzed were CP (%) and PPI (%). To determine the significant differences between the samples, the least significant difference Tukey test (LSD) was used at a significance level of 95%.

## 3. Results and Discussion

### 3.1. Color

All emulsions can be described by a pale-yellow color, according to a medium to high luminosity (Table 2).

No differences were found in luminosity regarding CP content; however, the highest contents of PPI (0.75 and 1%) led to significantly higher values of luminosity (*p* < 0.05). Luminosity increase of the emulsions might be explained by the increase of droplet concentration due to a higher light scattering through the droplets [21,22]. This increase in droplet concentration, obtained through the decrease in droplet size, will be observed in the samples elaborated with the highest contents of PPI in the microscopy section.

Concentration of CP and PPI did not affect color coordinate a*. However, increasing the % of CP, increased values of b* significantly (*p* < 0.05) regardless of PPI concentration. Increase of yellow color (+b*) could be due to the specific color of CP.

The color attribute C* measures the purity, saturation, or chroma of a color; a value of 0 is for a C* achromatic stimulus, with no orientation toward red, green, blue, or yellow; and a high value of C* means high saturation [23]. It should be considered that saturation depends on a* and b* values. As in b* values, higher concentrations of CP increased C* values significantly (*p* < 0.05).

Hue (h* value) is the color attribute measured by the angle respect to the origin 0°, defined in the position of a* positive and b* = 0, which represents a strictly red color. Following the anti-clockwise direction, at 90°, it would be a strictly yellow color; at 180°, a strict green; and a strict blue at 270°. According to the results obtained, all emulsions have a yellow hue bordering on green. By increasing the % CP from 0.75 to 1.25%, h* had significantly lower values (*p* < 0.05) closer to yellow and thus were less greenish. Furthermore, the addition of protein did not significantly affect h*.

The total color difference values (ΔE*) were appreciable by the human eye (ΔE* > 3) for all the samples regarding emulsion A except for emulsion B. Not only CP concentration but also PPI concentration influenced color difference values; emulsions with 1.25% of CP presented higher color differences regardless of the proportion of PPI.

### 3.2. Oil Loss

To evaluate physical stability of emulsions, oil loss was measured (Figure 1). Different concentrations of CP had no significant effects (*p* > 0.05) on the oil loss percentage. All emulsions without protein or with a low percentage (emulsions A, B, C, and D) had an oil loss between 60 and 70% regardless of the CP concentration. Data obtained from this parameter indicated that the coating of the oil droplets by the polysaccharides (cellulose and pectin) present in CP as well as its stabilizing effect was not entirely effective. When emulsions were centrifuged, the weak interfacial film coating oil droplets broke and led to the release of a high quantity of oil from the emulsion, as occurred in the emulsions prepared with whey protein isolate (WPI)-low methoxyl pectin (LMP) complexes [24]. The addition of more than a 0.5% of PPI (emulsions E, F, and G) caused a synergistic stabilizing effect between polysaccharides and protein, which gave lower values of oil loss (around 15%). This decrease could be related to a more efficient coating of oil droplets.

Vélez-Erazo et al. [15] studied oil loss in oil-in-water emulsions (60:40) prepared using PPI and different hydrocolloids as stabilizers. An oil loss between 2% and 17% was obtained depending on the hydrocolloids used, which agrees with the results obtained in this work.

### 3.3. Rheology

Rheology plays a crucial role in the physical behavior of emulsions, both in their appearance and structure and hence in their stability. Emulsions with different proportions of emulsifiers present different rheological behaviors [25], so it is expected that the different emulsions studied in the current work will present different rheological behaviors, as they contain different concentrations of CP and PPI. The viscoelastic properties of the emulsions were measured and the storage modulus (G’) and loss modulus (G’’) were determined. The G’ corresponds to the elastic response, and the G’’ is a measure of the viscous response of the material.

Stress sweeps of the different samples were conducted to determine the linear viscoelasticity range (LVR) of each emulsion, which comprises the stress interval for which the values of the viscoelastic functions do not vary significantly. For all emulsions, G’ was always greater than G’’ at low stresses (Figure 2A), and thus, elastic (solid) behavior predominated over the viscous one (liquid). At higher stresses, a crossing point between G’ and G’’ was observed related to structure deformation from a certain stress value; these structural rearrangements of the emulsion droplets in a high-stress situation have been described previously in HIPEs stabilized by gelatin particles by Tan et al. [26]. Stress values were observed at the crossover point, and as expected, the progressive addition of CP and PPI improved the yield stress from 10 to 25 Pa, presenting emulsions with higher concentrations of CP and PPI and a broader LVR, as also observed by Wijaya et al. [24].

All the emulsions exhibited a behavior similar to a solid because storage modulus (G’) was greater than loss modulus (G’’), as frequency sweeps revealed (Figure 2B). In addition, this behavior was more notable in the emulsions with higher CP and containing protein.

Both the concentrations of CP and PPI affected elastic and viscous moduli (Table 3). G’ was significantly higher (*p* < 0.05) when increasing CP concentration from 0.75% to 1% or 1.25% and when adding PPI, producing an increase in the dominant solid behavior. This increase may be related to the adsorption and accumulation of the particles at the water–oil interface and with the formation of a network structure of droplets, as observed below in microstructural studies (Section 3.4). Likewise, Zeng et al. [27] observed Pickering HIPEs prepared with gliadin/chitosan hybrid particles. Thus, emulsions with 1.25% of CP and with PPI (D, E, F, and G) showed more structuring. However, G’’ presented significantly higher values (*p* < 0.05) when the CP concentration used was the highest, thus indicating a higher viscous component. Protein addition also affected the value of G’’.

Results from stress and frequency sweeps suggested that the rheological properties of the emulsions were determined by the high viscosity brought about by the aqueous phase and by the ability of the CP and PPI to be adsorbed at the oil–water interface; therefore, emulsions with a predominant solid character at stress values below 10–25 Pa (depending on the emulsion) were attained. The solid behavior was also attributed to the high viscoelasticity of the protein-polysaccharide layer formed at the oil–water interface, as observed below in Section 3.4.

Zeng et al. [27] prepared HIPEs with corn oil (80–90%) and with hybrid particles of gliadin and chitosan as emulsifiers and obtained an emulsion with a yield stress of 70 Pa and a G’ of approximately 800 Pa, values much higher than those presented by the emulsions studied in this work. Huang et al. [28] analyzed the rheological properties of Pickering HIPEs stabilized by chitosan-caseinophosphopeptide nanocomplexes and obtained G’ values between 100 Pa and 200 Pa at 1 Pa and 1 Hz and a yield stress between 20 and 50 Pa, values closer to those obtained in this work.

The flow curves registered in all samples showed the typical shear-thinning flow behavior; a decrease in the apparent viscosity was seen when increasing the shear-rate (Figure 3). The increase of CP concentration had no significant effects on the apparent viscosity values at 10 s^−1^ (Table 3), but the addition of protein from a certain concentration (0.50%) caused a significant increase (*p* < 0.05), which was higher when PPI concentration reached 0.75% and 1%. This increase in viscosity would cause less oil loss and greater stability observed in samples elaborated with PPI.

However, these values were lower than those obtained by other authors for this emulsion. Wijaya et al. [24] studied the rheological properties of HIPEs stabilized by LMP and WPI complexes, and obtained apparent viscosities values between 30 and 60 Pa·s at a shear rate of 10 s^−1^ at 25 °C.

### 3.4. Microstructure

Microscopy allows studying the distribution and dimensions of the emulsion components as well as obtaining information about the stability of the system. In Figure 4, images of the emulsions obtained by light field (Figure 4(A1–G1)) and fluorescence (Figure 4(A2–G2)) microscopy can be seen. Fat was stained in red and biopolymers in green-yellow.

Sample A (0.75% CP) showed most of the oil in free form (Figure 4(A1,A2)) although there was a small proportion of fat presented as small globules. A part of the CP was located around the small fat globules, giving them a yellow color. By increasing the CP concentration to 1% (emulsion B), a greater emulsifying effect was achieved (Figure 4(B1,B2)), with CP at the interface, surrounding globules of heterogeneous shape and large size ranging from 3.4 × 10^3^ to 1.5 × 10^5^ µm^2^. The fat fraction was observed more compartmentalized than in the previous sample although still in free form. In emulsion C (1.25% CP), the emulsifying effect of the CP was higher than in emulsions A and B. Fat globules, while still large (areas ranging from 2.2 × 10^3^ to 9.6 × 10^4^ µm^2^), adopted a more homogeneous size and showed a polyhedral shape (Figure 4(C1,C2)).

Protein addition improved the structuring of the emulsions. Emulsion D (1.25% CP, 0.25% PPI), with a higher proportion of biopolymers, showed them forming an interface that surrounded the fat globules (Figure 4(D1,D2)). There was probably an interaction between the CP polysaccharides and the pea protein. Smaller fat globules (ranging from 1.1 × 10^3^ to 4.8 × 10^4^ µm^2^) were seen in emulsion D when compared to samples without PPI (emulsions A, B, and C). By increasing the PPI concentration to 0.5% (emulsion E), the emulsion was more structured, with oil globules ranging from 4.5 × 10^2^ to 4.5 × 10^4^ µm^2^. Biopolymers were located at the interface, surrounding the globules, and in the continuous phase, keeping them separated (Figure 4(E1,E2)). The CP-PPI system favored the compartmentalization of the globules and stabilized the emulsions. These results coincide with those obtained in the rheology studies because the samples with 0.5% PPI presented a viscosity higher than the emulsions without PPI. Emulsion F (0.75% PPI) showed a honeycomb structure (Figure 4(F1,F2)) with packed and mostly polyhedral oil globules, whose areas ranged from 3.1 × 10^2^ to 4.3 × 10^4^ µm^2^. The CP-PPI system acted by surrounding and stabilizing oil droplets. This sample presented viscosity values higher than those of emulsion E. In emulsion G (1% PPI), as in emulsion F, oil appeared in polyhedral, homogeneous, and packed droplets (Figure 4(G1,G2)) but with a smaller size (ranging from 1.4 × 10^2^ to 2.5 × 10^4^ µm^2^) than in the other emulsions; therefore, a greater stability was achieved.

According to the composition of emulsions and the images obtained, the increase of CP concentration from 0.75% to 1.25% favored the compartmentalization of the oil droplets in the emulsion. By increasing its concentration, the oil in the emulsion was shown as large and irregular droplets instead of being mainly free and unstructured. This may be due to a more effective coating of the oil droplets by the polysaccharides (cellulose and pectin) present in the CP, thus achieving greater stability.

Furthermore, emulsions with PPI presented a better packing of oil droplets than the emulsions without PPI, which also agrees with results obtained in rheology. From concentrations 0.5% PPI and above, there was a notable presence of smaller droplets shown by the decrease of the minimum and maximum area, which could be related to a lower loss of oil. This may be due to a synergistic stabilization effect occurring when the biopolymers of the CP-PPI system interact and thus a more efficient coating of the oil droplets.

Therefore, the greater stability attained in emulsions E, F, and G may be related to the protein-polysaccharide complexes at the oil-water interface, which formed a physical barrier on the surface of the oil droplets and enhanced their interaction among droplets. Furthermore, the presence of polysaccharides in the continuous phase may have helped to construct a network and increased the viscosity of the emulsion [25,28], thus acting as stabilizers.

## 4. Conclusions

This study elucidated if unmodified CP could be used as an emulsifier and stabilizer in HIPEs by itself or whether other ingredients (e.g., protein) are needed to improve the emulsion properties. PPI addition improved HIPEs physical stability, as shown in oil loss results of emulsions with 1.25% of CP and over 0.50% of PPI. Rheological measurements also showed more elastic behavior and higher apparent viscosity values when over 0.5% of PPI was added to HIPEs. A more efficient coating of the oil droplets was seen by microscopy when increasing CP and PPI concentrations due to the synergistic stabilization effect of CP and PPI system interactions.

The results showed that stable HIPEs with 1.25% of CP and PPI over 0.5% can be obtained; these HIPEs could be used to formulate emulsions for some food industries, such as mayonnaise, filling, or cream industries.

## Figures and Tables

**Figure 1 foods-11-00994-f001:**
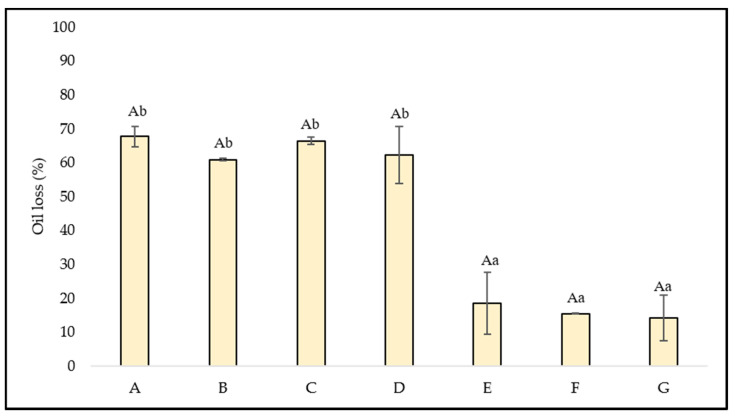
Oil loss (%) of the different emulsions. Means values with different capital letters differ significantly (*p <* 0.05) for CP (%). Means values with different lower-case letters differ significantly (*p <* 0.05) for PPI (%).

**Figure 2 foods-11-00994-f002:**
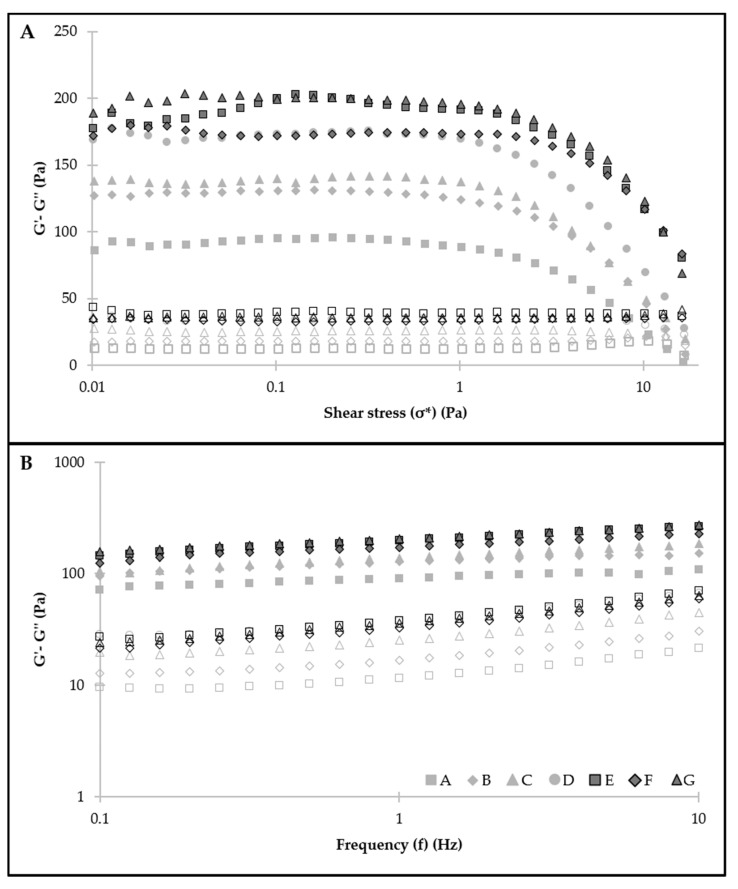
Stress sweep at 1 Hz (**A**) and frequency sweep at 0.3 Pa (**B**) of the different emulsions. Solid symbols represent G’, and open symbols represent G’’.

**Figure 3 foods-11-00994-f003:**
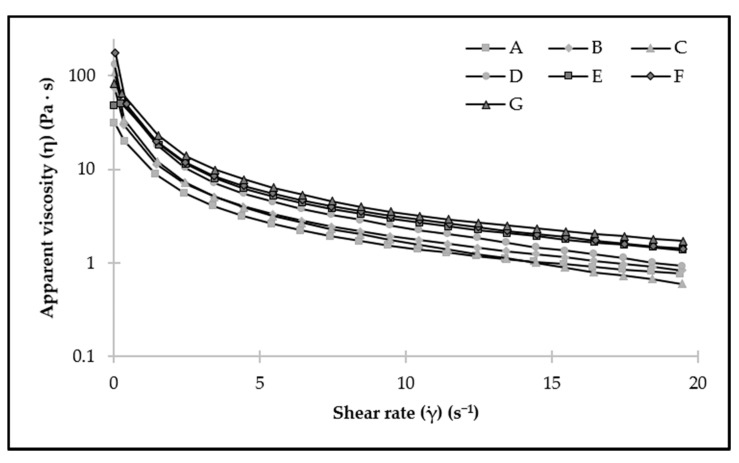
Flow curves of the different emulsions stabilized using CP and PPI.

**Figure 4 foods-11-00994-f004:**
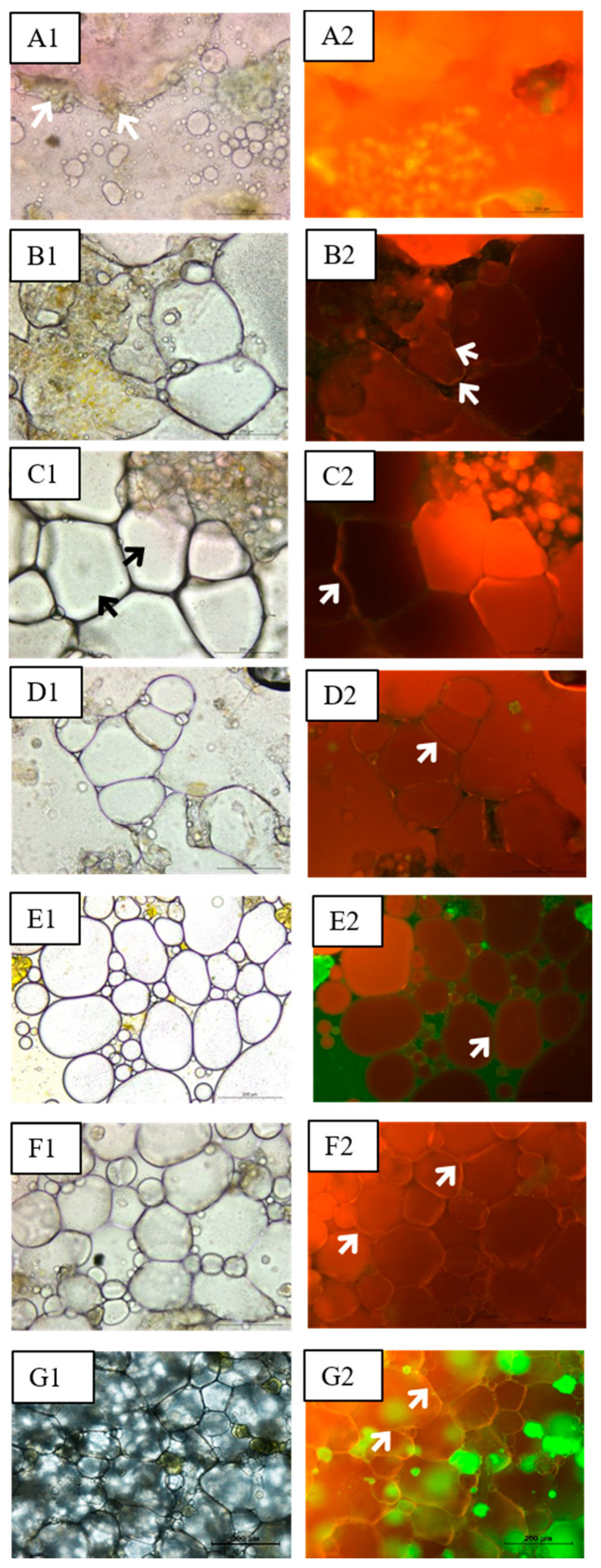
Images of light field microscopy (**A1**–**G1**) and fluorescence microscopy (**A2**–**G2**). Scale bar 200 µm; 10× magnification. White arrows mark citrus pomace, and black arrows mark polyhedral globules.

**Table 1 foods-11-00994-t001:** Emulsion formulation using different concentration of citrus pomace (CP) (%) and pea protein isolate (PPI) (%).

		PPI (%)				
		0.00	0.25	0.50	0.75	1.00
CP (%)	0.75	A				
	1.00	B				
	1.25	C	D	E	F	G

**Table 2 foods-11-00994-t002:** CIE L*a*b* color coordinates, color attributes (C* and h*) of emulsions, and total color differences related to A sample (ΔE*).

SAMPLE	L*	a*	b*	C*	h*	∆E*
A	58.30 ^Aa^ (2.04)	−4.58 ^Aa^ (0.21)	26.71 ^Aa^ (0.91)	27.10 ^Aa^ (0.93)	99.72 ^Ba^ (0.21)	-
B	57.57 ^Aa^ (1.16)	−4.46 ^Aa^ (0.45)	28.89 ^Ba^ (0.56)	29.24 ^Ba^ (0.52)	98.78 ^ABa^ (0.96)	2.31
C	57.74 ^Aa^ (3.42)	−4.10 ^Aa^ (0.35)	31.21 ^Ca^ (2.08)	31.48 ^Ca^ (2.08)	97.48 ^Aa^ (0.65)	4.57
D	60.58 ^Aab^ (1.28)	−4.08 ^Aa^ (0.69)	31.01 ^Ca^ (1.92)	31.28 ^Ca^ (1.83)	97.54 ^Aa^ (1.67)	4.90
E	60.76 ^Aab^ (0.30)	−3.88 ^Aa^ (0.16)	30.73 ^Ca^ (0.45)	30.98 ^Ca^ (0.43)	97.21 ^Aa^ (0.36)	4.77
F	62.24 ^Ab^ (2.16)	−4.11 ^Aa^ (0.46)	30.08 ^Ca^ (0.91)	30.36 ^Ca^ (0.85)	97.80 ^Aa^ (1.07)	5.21
G	61.26 ^Ab^ (0.62)	−3.56 ^Aa^ (0.08)	30.99 ^Ca^ (0.73)	31.20 ^Ca^ (0.73)	96.55 ^Aa^ (0.12)	5.31

The values are presented as a mean (standard deviation). Means values in a column with different capital letters in superscript differ significantly (*p <* 0.05) for CP (%). Means values in a column with different lower-case letters in superscript differ significantly (*p <* 0.05) for PPI (%).

**Table 3 foods-11-00994-t003:** Elastic (G’) and viscous (G’’) moduli values at 1 Hz, 0.3 Pa, and 20 °C. Apparent viscosity (η) at 10 s^−1^ and 20 °C.

Sample	G’ (Pa)	G’’ (Pa)	η (Pa·s)
A	90.62 ^Aa^ (4.63)	11.72 ^Aa^ (0.65)	1.32 ^Aa^ (0.05)
B	128.82 ^Ba^ (5.49)	17.32 ^Ba^ (0.73)	1.82 ^Aa^ (0.05)
C	137.37 ^Ba^ (10.56)	24.84 ^Ca^ (1.64)	1.58 ^Aa^ (0.01)
D	157.93 ^Bb^ (27.02)	33.54 ^Cb^ (3.52)	1.84 ^Aab^ (0.42)
E	181.90 ^Bc^ (17.27)	36.49 ^Cc^ (2.29)	2.43 ^Ab^ (0.29)
F	169.70 ^Bbc^ (5.82)	32.38 ^Cb^ (0.64)	2.84 ^Ac^ (0.06)
G	206.93 ^Bd^ (13.41)	37.10 ^Cc^ (2.71)	3.01 ^Ac^ (0.14)

The values are presented as a mean (standard deviation). Means values in a column with different capital letters in superscript differ significantly (*p* < 0.05) for CP (%). Means values in a column with different lower-case letters in superscript differ significantly (*p* < 0.05) for PPI (%).

## Data Availability

Not applicable.

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
