# Peer review of "High Internal Phase Emulsions Preparation Using Citrus By-Products as Stabilizers"

_foods, 2022, doi:10.3390/foods11070994_

Round 1
Reviewer 1 Report
General comments
The manuscript is a preliminary descriptive study about high internal phase emulsions (HIPEs) using citrus pomace (CP) as emulsifier and stabilizer with or without pea protein isolate (PPI) incorporation.
Many weaknesses are seemed from this study. First, the title indicates a “design” of the emulsions, which is not correct; instead, a series of formulations were done using different amount of CP, without any justification on the levels used. There is no characterization of the material used as emulsion stabilizer, which limit the possibility of making a rational design of the emulsions. Because of CP is an unknown material to fabricate HIPEs, a preliminary study about the emulsifying capacity of CP was be done to stablish the minimum amount of CP to be used in the fabrication of the HIPEs. Authors stated that pea protein has an amphiphilic nature, but most of the globular proteins are amphiphilic, hence, this fact is not a justification for the use of PPI. The aim of the work was to obtain HIPEs as potential replacers of plastic fats; however, no plastic fat was used as a control in this study to compare the properties of the CP-stabilized emulsions with a plastic fat. In the last paragraph of the discussion, authors talk about stability, but physical stability of the emulsions was not measured, which is critical in emulsion fabrication. Because of the lack of characterization of CP, discussions are not deep in terms of the mechanism of emulsions stabilization by CP and the possible molecular interaction between CP and PPI.
Specific comments
Line 46. How plastic fats extend the shelf life of foods?
Lines 70 – 71. Authors should justify the use of PPI instead other food proteins widely studied as emulsifiers.
Line 81. Raw material should be characterized. At least a proximal analysis should be done, as well as CP powder.
Lines 112 – 118. Measurement of emulsion color has no relation with the objective of the study. Authors should justify this measurement.
Equation 2. The symbol of percent (%) is missing after the number 100 to give it oil percent the corresponding unit.
Lines 124 – 126. Use subscripts to mass symbols of Equation 2.
Lines 134. Use superscript for the unit of shear rate.
Lines 143 – 144. Authors should indicate if the stain of the dispersed and continuous phases were done before or after emulsification.
Lines 148 – 149. Authors should indicate the procedure for measuring the area of the fat globules.
Table 2. The standard deviation of DE values is missing. Authors should incorporate it.
Figure 1. Decimal values in Y-axis should be eliminated.
Figure 2A. The name of the variable should be incorporated in the caption of the X-axis, the symbol is not enough.
Lines 234 – 235. The higher values of G’ compared to G’’ values are not enough evidence to claim that emulsions exhibited a behavior like a gel. Authors should go deep inside this discussion.
Table 3. Use superscript for the unit of shear rate.
Line 272. Use superscript for the unit of shear rate.
Figure 3. The name of the variable should be incorporated in the caption of the X-axis and Y-axis, the symbol is not enough. Besides, lines seem to go further than the maximum value of the Y-axis (100 Pa s). Authors should clarify this fact.
Line 281. The term “speed range” is not correct in this sentence, please modify. Use superscript for the unit of shear rate.
Line 287. Please correct the scientific notation used.
Line 307. Authors should revise the calculation of the fat globules area. Figure 4 shows globules lower than 200 mm in effective diameter; however, diameters of about 1000 to 6000 mm are needed to obtain values in the range presented in the manuscript, assuming fat globules as a perfect sphere. Besides, standard deviation is in the order of the median values, this should be reviewed too.
Author Response
Thank you for your helpful revision. We answer your comments below.
General comments
The manuscript is a preliminary descriptive study about high internal phase emulsions (HIPEs) using citrus pomace (CP) as emulsifier and stabilizer with or without pea protein isolate (PPI) incorporation.
Many weaknesses are seemed from this study. First, the title indicates a “design” of the emulsions, which is not correct; instead, a series of formulations were done using different amount of CP, without any justification on the levels used.
The title of the manuscript has been modified, including the word “preparation” instead of “design”. Levels of CP were chosen based on preliminary studies, where we tried to maximize the amount of CP in order to valorize it. In these preliminary studies, the amounts of. CP over 1.25% did not result in stable emulsions. We have reworded lines 109-110.
There is no characterization of the material used as emulsion stabilizer, which limit the possibility of making a rational design of the emulsions. Because of CP is an unknown material to fabricate HIPEs, a preliminary study about the emulsifying capacity of CP was be done to stablish the minimum amount of CP to be used in the fabrication of the HIPEs
Information about moisture, carbohydrates, fiber, protein, fat and ash content has been added (lines 83-84). We are working now on the technofunctional characterization of the material.
Authors stated that pea protein has an amphiphilic nature, but most of the globular proteins are amphiphilic, hence, this fact is not a justification for the use of PPI.
Pea protein was chosen over other globular proteins, such as soy protein, because it does not cause allergic problems. Justification for the use of PPI has been added (lines 73-74).
The aim of the work was to obtain HIPEs as potential replacers of plastic fats; however, no plastic fat was used as a control in this study to compare the properties of the CP-stabilized emulsions with a plastic fat.
Literature data from model plastic fats were used to compare results (lines 293-295). Comparison with a plastic fat as a control was not made because its results from the rheological characterization would differ so much from HIPEs produced in this work that the statistical analysis would not result in significant differences among our samples.
In the last paragraph of the discussion, authors talk about stability, but physical stability of the emulsions was not measured, which is critical in emulsion fabrication.
We are referring to physical stability measured by oil loss analysis.
Because of the lack of characterization of CP, discussions are not deep in terms of the mechanism of emulsions stabilization by CP and the possible molecular interaction between CP and PPI.
We take into consideration the reviewer’s comment and we are now working on a deep characterization of the CP, which will be included in future works.
Specific comments
Line 46. How plastic fats extend the shelf life of foods?
Plastic fats are usually composed of saturated fatty acids (SFA). Their properties are derived from their structure: the absence of double bonds makes SFA less susceptible to oxidation. The less oxidation, the greater stability; therefore, the shelf life of foods containing plastic fats is improved. This idea has been included in lines 47-48.
Lines 70 – 71. Authors should justify the use of PPI instead other food proteins widely studied as emulsifiers.
Justification of the use of PPI has been added (lines 73-74).
Line 81. Raw material should be characterized. At least a proximal analysis should be done, as well as CP powder.
Composition data on CP powder have been included in lines 83-84.
Lines 112 – 118. Measurement of emulsion color has no relation with the objective of the study. Authors should justify this measurement.
The color was measured to know to what extent CP and PPI amounts affect emulsion color.
Equation 2. The symbol of percent (%) is missing after the number 100 to give it oil percent the corresponding unit.
The symbol % is included in the first term of the equation 2.
Lines 124 – 126. Use subscripts to mass symbols of Equation 2.
Subscripts have been modified (lines 128-129).
Lines 134. Use superscript for the unit of shear rate.
The unit has been modified (line 138).
Lines 143 – 144. Authors should indicate if the stain of the dispersed and continuous phases were done before or after emulsification.
The stain was done after depositing the sample on the glass slide. This information has been added to the text (lines 147-148).
Lines 148 – 149. Authors should indicate the procedure for measuring the area of the fat globules.
The method has been added (lines 152-154).
Table 2. The standard deviation of DE values is missing. Authors should incorporate it.
The standard deviation of DE values is not usually given in scientific works. It is a parameter that indicates the global difference in the color of the samples; therefore, it is calculated from the averages of the parameters L*, a* and b*. The calculation of DE to provide a standard deviation (SD) could lead to error since there is no established criterion to know between which repetitions SD should be calculated. In this regard, (p<0.05) has been removed to avoid misunderstandings (lines 192-195).
Figure 1. Decimal values in Y-axis should be eliminated.
Figure 1 has been modified in the manuscript.
Figure 2A. The name of the variable should be incorporated in the caption of the X-axis, the symbol is not enough.
Figure 2A has been modified.
Lines 234 – 235. The higher values of G’ compared to G’’ values are not enough evidence to claim that emulsions exhibited a behavior like a gel. Authors should go deep inside this discussion.
“Gel” has been replaced by “solid” (lines 239-240). From a rheological perspective, an actual elastic gel network has been established when G’ is at least one order of magnitude greater than G,” and either modulus is not or is only slightly dependent on frequency. Therefore, our samples did not behave as gels since they did not fully fulfill these requirements.
Table 3. Use superscript for the unit of shear rate.
Unit has been modified (line 255).
Line 272. Use superscript for the unit of shear rate.
Unit has been modified (line 278).
Figure 3. The name of the variable should be incorporated in the caption of the X-axis and Y-axis, the symbol is not enough. Besides, lines seem to go further than the maximum value of the Y-axis (100 Pa s). Authors should clarify this fact.
Figure 3 has been modified. There was a mistake with Y-axis maximum value, and it has been corrected. Now, it is possible to see the start of the lines.
Line 281. The term “speed range” is not correct in this sentence, please modify. Use superscript for the unit of shear rate.
“Speed range” has been changed by “shear rate” (line 287).
Unit has been modified (line 287).
Line 287. Please correct the scientific notation used.
Scientific notation has been modified (line 293).
Line 307. Authors should revise the calculation of the fat globules area. Figure 4 shows globules lower than 200 mm in effective diameter; however, diameters of about 1000 to 6000 mm are needed to obtain values in the range presented in the manuscript, assuming fat globules as a perfect sphere. Besides, standard deviation is in the order of the median values, this should be reviewed too.
Thanks for noticing. Fat globules area has been calculated again and, indeed, there was a mistake in the results. Now, the results are given as the range of maximum and minimum area for each sample.
Reviewer 2 Report
The manuscript reported a method to obtain HIPEs with unmodified CP. Complimentary studies were done in rheology, microscopy etc. to support the finding. The study is sound and provides new scientific knowledge to the society.
Author Response
The authors thank the reviewer for the revision of the manuscript and the comments on it.
Reviewer 3 Report
Manuscript number: foods-1605260
Article Type: Article
Title: High internal phase emulsions design using citrus by-products as stabilizers
The Authors obtained high internal phase emulsions (HIPEs) as potential replacers of plastic fats, using unmodified citrus pomace (CP) as an emulsifier and stabilizer to valorize citrus by-products. They also evaluated protein isolate (PPI) to confer a higher stability to the emulsions. Color, oil loss, rheological behavior, and microstructure characteristics of the HIPEs were evaluated to understand their stability
Comments:
Lines 238-242: “Both the concentrations of CP and PPI affected elastic and viscous moduli (Table 3). G’ was significantly higher (p < 0.05) when increasing CP concentration from 0.75% to 1% or 1.25%, and when adding PPI, producing an increase in the dominant solid behavior. This increase may be related to the adsorption and accumulation of the particles at the water-oil interface and with the formation of a network structure of droplets”. You say that increase may be related to the adsorption and accumulation of the particles at the water-oil interface and with the formation of a network structure of droplets”. Could You tell what kind of measurements can prove it?
Figure 2A: What about points near 10 Pa on the x axis? I think is not necessary put them their, because is not important.
Figure 2B: Why do You say about 1Hz if You sweep plot between 0.1 – 10 Hz? In the Table 3, I see modulus at 1Hz, and its correct. Description of fig 2B should be correct to be clear
Lines 134, 272 etc.: for example: 10 s-1, please correct on 10 s-1.
Lines 254- 260: “Results from stress and frequency sweeps suggested that the rheological properties of the emulsions were determined by the properties of particles (CP and PPI) and by their ability to adsorb at the oil-water interface, therefore emulsions with a predominant solid character at stress values below 10–25 Pa (depending on the emulsion) were attained. This may be attributed to the high viscoelasticity of the protein-polysaccharide layer formed at the oil-water interface, and also due to the high viscosity brought about by the close packing of oil droplets” Again You are saying about: This may be attributed to the high (…). What You propose to prove it?
Recommendation: Reconsider after major revision (control missing in some experiments)
Author Response
Thank you for your helpful revision. We answer your comments below.
The Authors obtained high internal phase emulsions (HIPEs) as potential replacers of plastic fats, using unmodified citrus pomace (CP) as an emulsifier and stabilizer to valorize citrus by-products. They also evaluated protein isolate (PPI) to confer a higher stability to the emulsions. Color, oil loss, rheological behavior, and microstructure characteristics of the HIPEs were evaluated to understand their stability
Comments:
Lines 238-242: “Both the concentrations of CP and PPI affected elastic and viscous moduli (Table 3). G’ was significantly higher (p < 0.05) when increasing CP concentration from 0.75% to 1% or 1.25%, and when adding PPI, producing an increase in the dominant solid behavior. This increase may be related to the adsorption and accumulation of the particles at the water-oil interface and with the formation of a network structure of droplets”. You say that increase may be related to the adsorption and accumulation of the particles at the water-oil interface and with the formation of a network structure of droplets”. Could You tell what kind of measurements can prove it?
Microstructural study reveals that the particles are located in the interface surrounding fat globules. A network structure is also observed in microstructural studies when high concentrations of CP and PP are used. Comment on this idea has been included in lines 247-248.
Figure 2A: What about points near 10 Pa on the x axis? I think is not necessary put them their, because is not important.
Figure 2A has been modified.
Figure 2B: Why do You say about 1Hz if You sweep plot between 0.1 – 10 Hz? In the Table 3, I see modulus at 1Hz, and its correct. Description of fig 2B should be correct to be clear
Figure caption has been corrected (line 238).
Lines 134, 272 etc.: for example: 10 s-1, please correct on 10 s-1.
Units have been modified (lines 138, 255, 278, 287).
Lines 254- 260: “Results from stress and frequency sweeps suggested that the rheological properties of the emulsions were determined by the properties of particles (CP and PPI) and by their ability to adsorb at the oil-water interface, therefore emulsions with a predominant solid character at stress values below 10–25 Pa (depending on the emulsion) were attained. This may be attributed to the high viscoelasticity of the protein-polysaccharide layer formed at the oil-water interface, and also due to the high viscosity brought about by the close packing of oil droplets” Again You are saying about: This may be attributed to the high (…). What You propose to prove it?
We propose microstructural analysis for proving that droplets make a network and that the particles are located at the interface between water and oil. Comment on this has been included in line 266.
Round 2
Reviewer 3 Report
I accept manuscript in present form
Author Response
thanks